# Short-Course Regimen for Multidrug-Resistant Tuberculosis: A Decade of Evidence

**DOI:** 10.3390/jcm9010055

**Published:** 2019-12-25

**Authors:** Arnaud Trébucq, Tom Decroo, Armand Van Deun, Alberto Piubello, Chen-Yuan Chiang, Kobto G. Koura, Valérie Schwoebel

**Affiliations:** 1International Union Against Tuberculosis and Lung Disease, 75006 Paris, France; apiubello@theunion.org (A.P.); cychiang@theunion.org (C.-Y.C.); KGKoura@theunion.org (K.G.K.); vschwoebel@theunion.org (V.S.); 2Department of Clinical Sciences, Institute of Tropical Medicine, 2000 Antwerp, Belgium; tdecroo@itg.be; 3Research Foundation Flanders, 1000 Brussels, Belgium; 4Mycobacteriology Unit, Institute of Tropical Medicine, 2000 Antwerp, Belgium; avdeun@itg.be; 5Damien Foundation, POBox 1065, Niamey, Niger; 6Division of Pulmonary Medicine, Department of Internal Medicine, Wanfang Hospital, Taipei Medical University, Taipei 11696, Taiwan; 7Division of Pulmonary Medicine, Department of Internal Medicine, School of Medicine, College of Medicine, Taipei Medical University, Taipei 11031, Taiwan; 8COMUE Sorbonne Paris Cité, Université Paris Descartes, Faculté des Sciences Pharmaceutiques et Biologiques, 75006 Paris, France; 9École Nationale de Formation des Techniciens Supérieurs en Santé Publique et en Surveillance Epidémiologique, Université de Parakou, Parakou, Benin

**Keywords:** tuberculosis, treatment, MDR, 9-month regimen, outcome analysis, fluoroquinolones

## Abstract

About ten years ago, the first results of the so-called “Bangladesh regimen”, a short regimen lasting nine months instead of 20 months, revolutionized multidrug-resistant tuberculosis (MDR-TB) treatment. Similar short regimens were studied in different settings, relying for their efficacy on a later generation fluoroquinolone, either gatifloxacin, moxifloxacin, or levofloxacin. We review the published material on short MDR-TB regimens, describe their different compositions, their results in national tuberculosis programs in middle- and low-income countries, the risk of acquiring resistance to fluoroquinolone, and the occurrence of adverse events. With over 80% success, the regimen performs much better than longer regimens (usually around 50%). Monitoring of adverse events allows adapting its composition to prevent severe adverse events such as deafness. We discuss the current applicability and usefulness of the short injectable-containing regimen given the 2019 recommendation of the World Health Organization (WHO) for a new long all-oral regimen. We conclude that the most effective fluoroquinolone is gatifloxacin, currently not listed as an essential medicine by WHO. It is a priority to restore its status as an essential medicine.

## 1. Introduction

Since the late 1990s, a decade after the widespread use of rifampicin-throughout regimens, the increasing prevalence of resistance to rifampicin threatens TB control [1].The standard regimen (six months of rifampicin (R) and isoniazid (H) supplemented the two first months by pyrazinamide (Z) and ethambutol (E)—2RHZE/4RH) showed poor results in patients with rifampicin-resistant TB (RR-TB) [2]. New treatment regimens for multidrug-resistant/rifampicin-resistant tuberculosis (MDR/RR-TB) were needed. This was a very new field of research. The Green Light Committee (GLC) was established in 2000, with the purpose of enhancing access to quality-assured second-line drugs at competitive prices and ensuring that treatment was provided according to the World Health Organization (WHO) guidelines. The initial treatment phase commonly consisted of five drugs (including an injectable) and lasted for at least eight months, or six months past conversion; the second phase continued with the oral agents for a minimum of 12 months; thus with a minimum total treatment duration of 18 months. Although not standardized, in most cases, a fluoroquinolone (FLQ) and an injectable agent formed the basis of the regimen. The other drugs often included prothionamide (or ethionamide), P-amino salicylic acid (PAS), cycloserine, terizidone, pyrazinamide, and ethambutol [3].

The results of these long treatment regimens under program conditions were disappointing. The first WHO global report of MDR-TB outcomes was published in 2012 showing data from patients started on treatment in 2009. The overall treatment success was 48%; 10% had treatment failure; and 28% were lost to follow-up [4]. The same year (2012), an analysis of individual patient data showed very similar results [5]. Toxicity, long duration, and high costs were identified as causes of poor outcomes [4]. Moreover, a substantial proportion of unsuccessfully treated patients had XDR-TB (extensively drug-resistant TB due to strains not only resistant to rifampicin but also to FLQs and aminoglycosides) [4].

Facing these very unsatisfactory results, Van Deun and his team in Bangladesh looked for a shorter and more effective treatment regimen to minimize treatment failure and lost to follow-up, and to standardize MDR/RR-TB management. After evaluating several combinations of drugs and treatment durations, a nine-month regimen was identified as the most effective [6]. This regimen included high-dose gatifloxacin (Gh), clofazimine (C), ethambutol (E), and pyrazinamide (Z) throughout, supplemented by prothionamide (P), kanamycin (K), and high-dose isoniazid (Hh) during an intensive phase of a minimum of six months: 4–6 KHhPGhCEZ/5 GhCEZ. The intensive phase was extended by monthly increments up to six months, keeping the duration of the continuation phase at five months. Therefore, the overall treatment duration could be up to 11 months in patients without sputum smear conversion at four months This shorter treatment regimen (STR) resulted in 87.9% (95% confidence interval, 82.7–91.6) relapse-free cure among 206 patients. Surprisingly, the publication of these excellent results was received with a lot of skepticism. Success was attributed to the specific Bangladeshi setting. In 2014, the same team published an update showing very similar results for 515 patients [7]. 

Meanwhile the STR was piloted in multiple settings. In this manuscript, we summarize the findings obtained from these settings, including the composition of the different regimens, their outcomes, the risk of acquiring resistance to FLQs, and the occurrence of adverse events. Moreover, we discuss how evidences on the STR informed successive World Health Organization (WHO) guidelines and are challenging the 2019 WHO recommendations.

## 2. Methodology

A review of relevant published literature on cohorts treated with the Bangladesh STR or STR with minor modifications was performed. To identify relevant citations, we searched Pubmed using strings that combined the following search terms: “multidrug-resistant tuberculosis”, “rifampicin-resistant tuberculosis”, “outcome”, “short-course”. Additional citations were identified from the reference lists of retrieved citations. Studies not showing original research findings were excluded, as well as studies not showing treatment outcomes.

Of the citations retrieved, we describe the design, the composition of the different STR studied, and report on their outcomes following the authors’ definitions. Most authors used the WHO definitions except for minor adjustments, with the following main categories: success (either cure or completion), lost to follow-up, death, treatment failure, and relapse [8]. We define treatment failure and relapse as bacteriologically unfavorable outcomes; treatment failure, relapse, death, and lost to follow-up as programmatically unfavorable outcomes. We also report on numbers of acquired resistance to FLQ per 1000 patients initially susceptible to FLQ and treated with an STR.

If not presented in the publications, proportions and 95% confidence intervals (95% CI) were calculated using Stata version 14.2.

## 3. Results

Six publications reporting on latest original findings of studies conducted between 2005 and 2015 in Asia and Africa were identified. Table 1 shows the designs of the different STR studies. Most studies were prospective cohort studies. One study was a randomized clinical trial. 

### 3.1. Composition of the Initial Regimen

The rationale for the composition of the Bangladesh STR was published previously [6]. Gatifloxacin is the core drug of this regimen, the one drug that contributes most to relapse-free cure of TB [14]. The combination of the high early bactericidal activity and the high sterilizing effect (i.e., the power to eliminate persistent bacilli) of this fourth-generation FLQ allowed shortening of the treatment duration from 20 to 9 months [15,16]. It was used at high dosage, taking into account experimental evidence of suppression of resistant mutants [17]. The other drugs in the regimen are companion drugs, of which some are more important than others. Kanamycin is used for its early bactericidal activity, killing actively dividing bacilli, thus reducing the risk of selecting resistant mutants leading to treatment failure [3]. Pyrazinamide and clofazimine reduce the risk of relapse by their sterilizing activity, thus eliminating bacilli with low metabolic activity [18]. Isoniazid, ethambutol, and prothionamide are included mainly for additional protection of the core drug. High-dose instead of normal-dose isoniazid is preferred because a higher concentration has been shown to overcome low-level resistance and may be particularly useful for patients with thioamide cross-resistance [19,20]. Prothionamide is given only during the intensive phase to limit frequent gastrointestinal adverse events, often responsible for loss to follow-up. 

### 3.2. Overall Efficacy of the Different STRs

In 2008, Benin and Cameroon, inspired by the preliminary results of the Bangladesh STR, launched a study of a slightly modified STR: the continuation phase lasted eight instead of five months, with a normal instead of a high dose of gatifloxacin, and with prothionamide throughout [9,11]. 

In 2008, Niger started using a 12-month STR similar to Benin/Cameroon, but with high-dose gatifloxacin and without prothionamide in the continuation phase [10]. 

In 2013, a large study was launched in nine countries in West and Central Africa (called the Nine-country study) to evaluate the original Bangladesh nine-month regimen, but with normal-dose moxifloxacin (Mfx) instead of high-dose gatifloxacin [12,21].

For all these regimens, the duration of the continuation phase was fixed, but the four-month intensive phase could be extended for a maximum of two months if sputum smear microscopy showed bacilli at the end of the fourth month. 

Outcomes are shown in Table 2 and Figure 1. Treatment success varied between 80.2% and 95.5%. Lost to follow-up, death, treatment failure, and relapse varied between 0% and 7.8%, 4.5% and 9.2%, 0% and 5.9%, and 0% and 3.3%, respectively. To detect eventual relapses post-treatment, follow-up was at least one year and about two years or more for Bangladesh, STREAM, and the West/Central Africa studies.

Finally, the STREAM trial randomized 383 participants to receive a STR (9–11 months) or a long 20-month individualized regimen following the 2011 WHO guidelines. The STR differed from the original Bangladesh regimen only by the substitution of high-dose gatifloxacin by high-dose moxifloxacin. The 2019 publication showed non-inferiority of the STR in persons with rifampicin-resistant but FLQ- and aminoglycoside-susceptible TB [13].

### 3.3. Initial Resistance

#### 3.3.1. Initial Resistance to FLQ

In Bangladesh, Niger, and in the Niger-country study, patients received the STR without knowing at initiation of treatment whether their tuberculosis was resistant or not to FLQ. Patients with a strain retrospectively found resistant to FLQ at the initiation of treatment were much more likely to have a programmatically unfavorable outcome (≥50%) than those with a susceptible strain (<20%) (Table 3). 

A recent study compiled data from Bangladesh, Niger, and Cameroon [23]. Among patients with initially FLQ-susceptible TB, low-level resistance, or high-level resistance to FLQ, 98.7%, 83.5%, and 57.4% had a bacteriologically favorable outcome, respectively. Low-level (vs. FLQ-susceptible; adjusted odds ratio (aOR) 16.0 (37.9–6.8)) and high-level initial resistance (aOR 122.1 (343.4–47.9)) to FLQ were associated with a bacteriologically unfavorable outcome (treatment failure or relapse). 

#### 3.3.2. Initial Resistance to Companion Drugs

In three prospective studies (Bangladesh, nine-country study, Niger), no correlation was found between initial resistance to pyrazinamide or ethambutol or prothionamide in patients with FLQ-susceptible TB (Table 4) [7,10,12].

In the STREAM trial, a bacteriologically unfavorable outcome was more likely with the STR in the presence of pyrazinamide resistance in the per protocol analysis, but not in the modified intention-to-treat analysis [13].

### 3.4. Amplification of Resistance to FLQ

Amplification of resistance can only be assessed in patients with failure or relapse. Table 5 presents data on acquisition of FLQ resistance in original studies among patients with initially documented FLQ-susceptible tuberculosis treated with an STR regimen. 

Cumulating data of patients from Bangladesh, Niger, and Cameroon, at most 1/859 patients with a gatifloxacin-based STR might have acquired resistance to FLQ, whereas for patients treated with a high-dose levofloxacin-based regimen 9.9 per 1000 acquired FLQ resistance, and for those treated with a moxifloxacin-based STR 17.5 per 1000 acquired resistance to FLQ [23]. 

In the nine-country and in the STREAM studies, patients were treated with a moxifloxacin-based STR, and respectively 14.0 per 1000 and 20.3 acquired resistance to FLQ [13,21]. 

In summary, there has been no proven amplification of resistance to FLQ when gatifloxacin was used, while there were significantly more cases of amplification with moxifloxacin or levofloxacin treatment, even at high dose.

### 3.5. Effect of HIV on Outcomes

The composition of the treatment regimen for MDR-TB does not differ for people living with HIV [24]. 

HIV is rare in Bangladesh and in Niger. Data on treatment outcomes according to the HIV status of the patients are available for the Cameroon and the nine-country study, where respectively 20.0% and 19.9% of patients were HIV positive (Table 6) [17,19]. Deaths were more frequent among the HIV positive than among the HIV-negative in the nine-country study (19.0% vs. 5%), and in Cameroon (10.0% vs. 5.8%). The frequency of failure, lost to follow-up, and relapse did not differ significantly according to HIV status.

In the STREAM study, 34% patients were HIV positive; 17.5% of the HIV positive died vs. 4.0% of the HIV negative. The other outcomes according to the HIV status are not available [13].

### 3.6. Adverse Events (AEs)

All MDR-TB treatments regimens provoke more or less severe AEs, and WHO recommends the implementation of pharmacovigilance and the collection of information on aDSM (active drug safety monitoring and management) of AEs [25]. Methods for defining and monitoring AEs differ, but most studies use international scales to grade the severity.

In terms of safety, the STREAM study reported that the STR was similar to the long regimen: 48.2% of Grade 3–5 adverse events for the STR against 45.4% for the long treatment, and respectively, 32.3% and 37.6% serious adverse events [13]. 

Gastro-intestinal disorders are the most frequent complaints reported in all the studies. Even if these events are not life threatening, they are an important cause of abandoning treatment. Although almost all the drugs can provoke nausea and vomiting, the thioamide (either prothionamide or ethionamide) is the main cause. When building the STR, this was a strong argument for using a thioamide only during the intensive phase of treatment.

Hearing deficiencies due to kanamycin are a serious problem [12,22,26]. Longer duration and cumulative dose are, besides old age, the only risk factors for ototoxicity shown in pharmacokinetic studies [27]. The STR uses this drug for 4–6 months, while the formerly recommended long WHO regimen was using it for a minimum of eight months, often longer. Hearing loss (any grade) is reported in 20%–40% of patients, and up to 2.6% turned deaf in the nine-country study [12]. In Niger, with countrywide use of STR including kanamycin, audiometry surveillance is rigorously organized since 2016, and linezolid replaces kanamycin as soon as hearing impairment is detected. Since the implementation of this measure, no patient treated with STR developed severe hearing loss [22]. 

The QT interval on the ECG represents the electrical depolarization–repolarization of myocardial cells that leads to ventricular contractions. Moxifloxacin and clofazimine are suspected of increasing the risk of QT prolongation [28]. QT prolongation was well documented in the STREAM trial and was not significantly more frequent with the use of the STR than with the long treatment (11.0% vs. 6.4%, *p* = 0.14) [13]. QT prolongation leading to severe clinical disorders, such as torsade de pointes is rare [28]; none was documented in the nine-country or the STREAM studies.

Gatifloxacin was struck from the WHO essential medicines list and banned in most countries, because of sometimes fatal dysglycemia incidents in elderly Canadian patients [29]. This was not seen in any of the Gfx-using countries mentioned after treating altogether over 1000 patients. Occasional hyperglycemia was easily managed, and otherwise these patients were successfully switched to 15 months ofloxacin STR [6].

### 3.7. Cost

Replacing kanamycin (K) by amikacin (A) according to the last WHO recommendations [24], and using the Global Drug Facility prices, the regimen 4 AHhPMhCEZ/5 MhCEZ costs 782 USD, much less than the long all-oral regimen currently recommended by the WHO (6 Bdq–Lfx–Lzd–Cfz–Cs/14 Lfx–Lzd–Cfz–Cs) (Bdq = bedaquiline, Lfx = levofloxacin, LZD = linezolid, CS = cycloserine)—6000 USD [30].

### 3.8. Sustainability of the STR

The STR cohort studies reported in this paper were implemented in national routine TB programs. Since the 2016 WHO recommendations [31], many national guidelines recommend the STR in patients with (likely) FLQ-susceptible RR-TB [32]. According to WHO by the end of 2018, 82 countries used the STR [33]. Bangladesh, Benin, Cameroon, and Niger, the first countries which implemented this regimen, are maintaining high cure rates (>80%) after more than 10 years of programmatic STR implementation [34]. 

## 4. Discussion

Implementation of the STR resulted in over 80% programmatic MDR-TB treatment success in 14 different countries in Africa and Asia. This contrasts with the 56% treatment success stated in the last 2019 WHO global tuberculosis report among patients under the long treatment regimen [33]. Moreover, it is well documented that irregular intake of anti-TB drugs in insufficient or inappropriate dose, due to frequent adverse events and long treatment duration, favors the development of XDR, thus explaining the rapid worldwide expansion of the XDR epidemic following the use of the longer treatment regimen [4,35].

In its 2016 guidelines, relying on findings of STR cohort studies, WHO recommended: “In patients with rifampicin-resistant or multidrug-resistant TB, who have not been previously treated with second-line drugs and in whom resistance to fluoroquinolone and second-line injectable agents has been excluded or is considered highly unlikely, a shorter MDR-TB regimen of 9–12 months may be used instead of a conventional regimen” [31].

In its 2019 guidelines, WHO still recommends the STR. However, relying on an individual patient data meta-analysis assessing the effect of individual drugs on treatment outcomes [36], a long all-oral regimen is preferred (conditional recommendation, very low certainty in the estimates of effect) [24]. If there is no contra-indication, this new long regimen should include at least bedaquiline, a FLQ, linezolid (strong recommendation, moderate certainty in the estimates of effect), complemented with a fourth drug. To date, not a single publication documented the bacteriological outcomes of this recommended long all-oral regimen. Despite expected improved efficacy, failure or relapse will undoubtedly occur, as no regimen has ever proved 100% effective in TB history. The proposed combination may jeopardize treatment options for those patients, particularly if failure strains are resistant to both second-line core drugs: bedaquiline and FLQ [37]. Indeed, some small studies of bedaquiline-containing regimens show high rates of acquired drug resistance to bedaquiline: 24 out of 116 tests in Russia [38], and 6 out of 30 in Pakistan [39]. An alternative approach, applying the cascade of regimens concept, seems a better option: in TB treatment regimens, the core drug should be rifampicin when the strain is susceptible to rifampicin, a later generation FLQ in case of resistance to rifampicin, and bedaquiline if there is a resistance to FLQ [14]. 

Although not recommended in the current version of the WHO MDR-TB guidelines, the most effective FLQ is gatifloxacin. When high-dose gatifloxacin is used, low-level resistance can be overcome [23,40]. When another FLQ than gatifloxacin was used as STR core drug, between 10 and 20 per 1000 with initially FLQ-susceptible TB acquired resistance to FLQ. That is a real public health danger. In comparison, acquired resistance to rifampicin after the six-month first-line regimen is in the order of 1 per 1000 [41]. To not lose the second step of the treatment cascade, high-dose gatifloxacin should be the FLQ of choice. However, presently there is no quality-assured preparation of gatifloxacin on the market, and this drug cannot be bought by the countries through the Global Fund. Its reintroduction on the Essential drug list of the WHO is crucial for the future of effective and scalable MDR/RR-TB treatment [40,42].

Data from the studies reported here show that initial resistance to pyrazinamide or a thioamide or ethambutol are no contraindications for the STR. This contrasts with the 2019 WHO guidelines, which recommends to not prescribe the STR in patients with strains resistant to pyrazinamide or ethionamide/prothionamide [24]. These restrictions come mainly from the already-cited individual patient data meta-analysis showing a higher risk of treatment failure and relapse in patients with such strains compared to susceptible strains [36]. Interestingly, a previous individual meta-analysis showed that the inclusion of ethionamide/prothionamide as well as a longer use of injectables in the regimen were associated with treatment success; clofazimine, now considered as an important companion drug, was found not to have significant activity [5]. Several issues arise from the manner in which such meta-analyses were performed, as these assess the efficacy of individual drugs, neglecting the regimen in which individual drugs were used. Decades of TB research has shown that the rationale for the composition of a regimen goes far beyond combining a given number of likely active drugs [43]. Hence, the bacteriological activity of other drugs and the overall strengths of the treatment regimen cannot be ignored when assessing the effect of an individual drug on TB treatment outcomes. 

Hearing loss linked to the use of injectables is of serious concern but is usually manageable if surveillance of audition is regularly performed and the drug stopped on time. The replacement of aminoglycoside by linezolid in case of initial hearing deficiency or of hearing diminution during treatment has been proven possible, even in very-low-income countries. However, linezolid provokes adverse reactions which can be very serious and requires a careful monitoring and management of adverse events which is feasible to offer in low- and middle-income countries only for few selected patients but not in routine practice [22]. The STR is not contraindicated for HIV-positive patients [24], but hearing loss is more frequent among them and surveillance of audition must be reinforced [12,44].

WHO degrades the injectables, as the individual patient data meta-analysis showed that its use was associated with having a higher mortality, ignoring its association with prevention of acquired fluoroquinolone resistance [36]. If an injectable should still be included, the 2019 guidelines recommend using amikacin instead of kanamycin, for instance in the STR [24]. However, the comparison between both injectables may be biased. Being administered intravenously, amikacin was more frequently used in high-income countries (with high-quality supportive care), while intramuscular kanamycin was mainly used in low-income countries (with little supportive care). Moreover, within low–middle income countries, kanamycin was not associated with worse outcomes [36]. This recommendation is therefore disputed [45]. When using an injectable, it is recommended by pharmacologists to give it thrice weekly at the normal 15 mg/kg dose for only four months. That would also make the painful injections more acceptable to patients, while remaining equally effective [46,47].

The low cost of the STR favors sustainability in low-income countries, as Global Fund priorities and resources may shift over time.

Our review has some important limitations. Data on acquired fluoroquinolone resistance, adverse events, and HIV co-infection were not systematically collected in all settings. Data on initial DST results were incomplete in some settings, particularly for pyrazinamide, prothionamide, and ethambutol. On the other hand, studies reviewed included the single clinical trial on MDR-TB treatment ever completed, and the observational cohort studies were all followed by national roll-out which used the same data collection tools and procedures. Our findings based on program data are thus likely to reflect the reality of the management of MDR-TB in the routine of several national MDR-TB programs. 

## 5. Conclusions

Large observational studies, one randomized clinical trial, and country-level experiences have demonstrated the very high programmatic and bacteriological effectiveness, safety, and scalability of the STR, even in low-income countries. Adverse drug reactions in STR occur but remain manageable without treatment interruption. Standardization of treatment is essential for low- and middle-income countries to allow a regular drug supply and to decentralize services where human resources are scarce. 

For the near future, it would be of the utmost importance to restore gatifloxacin’s status as an essential medicine, as it is the first-choice core drug for the treatment of MDR-TB. It would be a real progress to abandon the use of injectable drugs but so far there is very weak evidence for the efficacy of injection-free regimens, and we recommend waiting for the results of ongoing studies. 

## Figures and Tables

**Figure 1 jcm-09-00055-f001:**
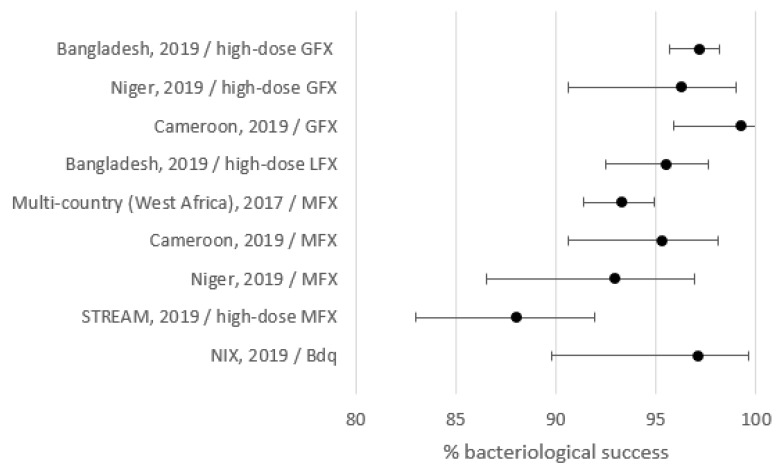
Treatment success by setting. GFX: gatifloxacin; MFX: moxifloxacin.

**Table 1 jcm-09-00055-t001:** Characteristics of published studies evaluating short-course regimens.

Study, Year	Country	Design	Shorter-Course Regimen
Aung, 2014 [7]	Bangladesh	Cohort, prospective	4–6 KCGhEHhZP/5 GhEZC
Gninafon, 2012 [9]	Benin	Cohort, retrospective	4–6 KCGEHhZP/8 GhEZCP
Piubello, 2014 [10]	Niger	Cohort, prospective	4–6 KCGhEHhZP/8 GhEZC
Kuaban, 2015 [11]	Cameroon	Cohort, prospective	4–6 KCGEHZP/8 GEZCP
Trébucq, 2018 [12]	Nine countries West and Central Africa ^$^	Cohort, prospective	4–6 KCMEHhZP/5 MEZC
Nunn, 2019 [13]	Ethiopia, South Africa, Mongolia, Vietnam	RCT (STREAM) ^#^	4–6 KCMhEHhZP/5 MhEZC

^$^ Benin, Burkina Faso, Burundi, Cameroon, Central Africa Republic, Cote d’Ivoire, Democratic Republic of Congo, Rwanda, Niger. ^#^ RCT = Randomized clinical trial. Patients were randomized to either the short course or the locally used standard long regimen. K—kanamycin; C—clofazimine; Gh—gatifloxacin; E—ethambutol; Hh—high-dose isoniazid; Z—pyrazinamide; P—prothionamide.

**Table 2 jcm-09-00055-t002:** Outcomes reported in published studies on short-course regimens.

Setting	Core Drug	Study Pop	Failure *n* (%)	LTFU ^a^ *n* (%)	Death *n* (%)	Relapse *n* (%)	Success *n* (%)	(95% CI)
Bangladesh	GFX ^h^	515	7 (1.4)	40 (7.8)	29 (5.6)	4 (0.8)	435 (84.5)	(81.0,87.5)
Benin	GFX	22	0	0	1 (4.5)	0	21 (95.5)	(77.2,99.8)
Niger	GFX ^h^	65	0	1 (1.5)	6 (9.2)	0	58 (89.2)	(79.0,95.6)
Cameroon	GFX	150	1 (0.7)	5 (3.3)	10 (6.7)	0	134 (89.3)	(83.3,93.8)
West/Central Africa ^$^	MFX	1006	59 (5.9)	48 (4.8)	78 (7.8)	14 (1.4)	807 (80.2)	(77.6,82.6)
STREAM ^£^	MFX ^h^	245	14 (5.7)	6 (2.4)	19 (7.8)	8 (3.3)	198 (80.8)	(75.3,85.6)

*n* = Number. GFX: gatifloxacin; MFX: moxifloxacin; the “h” after the name of the drug means high dosage. ^a^ LTFU = lost to follow-up. 95% CI—95% confidence interval. ^$^ Benin, Burkina Faso, Burundi, Cameroon, CAR, Cote d’Ivoire, DRC, Rwanda, Niger. ^£^ Data from modified intention-to-treat analysis was adapted to fit programmatic definitions.

**Table 3 jcm-09-00055-t003:** Association between initial resistance to fluoroquinolone and programmatically unfavorable outcomes.

		Programmatically Unfavorable *n* (%)	Total Tested	OR (95% CI)
Bangladesh [7]				
	FLQ susceptible	59 (13.4)	439	1
	FLQ high-level resistance	15 (51.7)	29	6.9 (3.2–15.0)
West/Central Africa [12,21] ^#^				
	FLQ susceptible	112 (19.6)	571	1
	FLQ resistance	12 (44.4)	27	3.31 (1.5–7.2)
Niger [10,22]				
	FLQ susceptible	32 (15.0)	214	1
	FLQ resistant	9 (69.2)	13	12.8 (3.7–44.1)

*n* = Number. OR = odds ratio; CI = confidence interval; FLQ = fluoroquinolone. ^#^ Benin, Burkina Faso, Burundi, Cameroon, CAR, Cote d’Ivoire, DRC, Rwanda, Niger.

**Table 4 jcm-09-00055-t004:** Association between initial resistance to companion drugs and programmatically unfavorable outcome among patients with initially fluoroquinolone-susceptible tuberculosis (TB).

	Setting	Programmatically Unfavorable *n* (%)	Total Tested	OR/RR 95% CI
Bangladesh				
	Prothionamide susceptible	63 (16.0)	394	1
	Prothionamide resistant	11 (13.3)	83	0.93 (0.42–1.9)
	Pyrazinamide susceptible	16 (11.1)	144	1
	Pyrazinamide resistant	18 (17.6)	102	1.1 (0.47–2.7)
West/Central Africa ^#^				
	Prothionamide susceptible	13 (17.6)	74	1
	Prothionamide resistant	31 (27.0)	115	1.73 (0.84–3.58)
	Pyrazinamide susceptible	37 (20.9)	177	1
	Pyrazinamide resistant	30 (17.3)	173	0.79 (0.46–1.36)
	Ethambutol susceptible	5 (17.2)	29	1
	Ethambutol resistant	16 (24.6)	65	1.57 (0.51–4.79)
Niger				
	Prothionamide susceptible	20 (11.9)	168	1
	Prothionamide resistant	6 (15.0)	40	1.3 (0.5–3.5)
	Pyrazinamide susceptible	18 (16.2)	111	1
	Pyrazinamide resistant	8 (8.6)	93	0.5 (0.2–1.2)
	Ethambutol susceptible	8 (12.8)	63	1
	Ethambutol resistant	18 (12.4)	145	1.3 (0.5–3.5)

*n* = Number. OR = odds ratio; CI = confidence interval. ^#^ Benin, Burkina Faso, Burundi, Cameroon, CAR, Cote d’Ivoire, DRC, Rwanda, Niger.

**Table 5 jcm-09-00055-t005:** Acquired resistance to fluoroquinolone among initially susceptible cases ^#^.

Setting, Core Drug	Initially Susceptible (A)	DST not Performed	Susceptible ^£^	Acquired Resistance (B)	Acquired Resistance per 1000
					(B/A)*1000
Bangladesh, Niger, Cameroon, GFX [23]	859	1	2	0	0
Bangladesh, Niger, Cameroon, LFX [23]	203	0	3	2	9.9
Bangladesh, Niger, Cameroon, MFX [23]	228	4	1	4	17.5
West/Central Africa ^$^, MFX [12]	571	25	8	8	14.0
STREAM, MFX h [13]	246	NA	NA	5	20.3

NA: data not available. GFX: gatifloxacin; MFX: moxifloxacin; LFX: levofloxacin: the “h” after the name of the drug means high dosage. ^#^ Recurrence: either treatment failure or relapse. ^$^ Benin, Burkina Faso, Burundi, Cameroon, CAR, Cote d’Ivoire, DRC, Rwanda, Niger. ^£^ Patients who failed or relapsed with strains still susceptible to fluoroquinolone.

**Table 6 jcm-09-00055-t006:** Outcomes according to the HIV status among patients treated in the Nine-country and the Cameroon studies.

	Nine-Country Study		Cameroon	
	HIV pos	HIV neg	HIV pos	HIV neg
	*n* (%)	*n* (%)	*n* (%)	*n* (%)
Success	145 (72.5)	664 (82.4)	25 (83.3)	109 (90.8)
Failure	9 (4.5)	51 (6.3)	0	0
Died	38 (19.0)	40 (5.0)	3 (10.0)	7 (5.8)
Lost to follow-up	8 (4.0)	37 (4.6)	2 (6.7)	4 (3.3)
Relapse	0	14 (1.7)	0	0
Total	200	806	30	120

pos = positive; neg = negative.

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
