# Peer review of "Short-Course Regimen for Multidrug-Resistant Tuberculosis: A Decade of Evidence"

_jcm, 2019, doi:10.3390/jcm9010055_

Round 1

Reviewer 1 Report

This is a good review of the evidence of the efficacy of the shourt-course regimen (SCR) for the treatment of multidrug-resistant tuberculosis (MDR-TB), including an injectable second-line drug, as opposed to the new all-oral treatment proposed by WHO. The authors present the arguments for keeping the SCR as a cheap and efficient option, particularly in countries with limited resources. Furthermore, they present the evidence demonstrating that gatifloxacin is the most efficient quinolone and should be re-introduced on the market.

An interesting point is also the fact that, according to the authors, the presence of resistance against some of the companion drugs (pyrazinamid, thioamides and ethambutol) has no impact on the final outcome of treatment. This is in contradiction with the current position of WHO, which proposes the new oral treatment as a method for overcoming these additional resistances and as an argument against the SCR.

Main comments:

1. The main issue is the fact that there is no consensus or evidence for the use of SCR in regions with the highest proportion of MDR-TB, namely Eastern Europe, where additional resistance to several second-line drugs is current and the unabated transmission of resistant strains represents a major threat for public health. The authors use the evidence from the meta-analysis of Ahuja (2012) for defending the use of injectables, ethambutol, thioamides and pyrazinamid but do not consider the fact that, in this analysis, a large proportion of patients did not receive quinolones and the results probably do not represent the current reality.

2. The other point which merits a more detailed discussion is the position of injectable drugs in the treatment regimens. It would be of interest to know the opinion of the authors about the recent analysis by Reuter A et al (IJTLD 2017;21(11):1114) listing all arguments against the use of injectable drugs and the deficient evidence for their efficacy.

3. As the authors rightly mention, there is currently only weak evidence of the efficacy of injection-free regimens for the treatment of MDR-TB and the included drugs are much more expensive. If, as can be expected, the evidence comes and the price of bedaquiline, linezolid and delamanid sinks, would the authors still claim that the addition of an injectable drug is necessary?

Minor comments:

Fig 1: please clarify the abbreviations GFX, GFXh, MFX and MFXh in the legend  Line 185 amplification of resistance: can probably be assessed only in patients wth relapse. Please clarify.  Line 274, discussion. Sequential addition of single drugs and insufficient dosage / serum levels of drugs with low bioavailability is probably much more important for the development of resistance than intermittent use of drugs in correct dosage. Suggest to add "..intake of anti-TB drugs in insufficient or  inappropriate dose… favour the development of XDR". I also suggest to add the reference from Shin SS, AJRCCM 2010:182:426-432 as an argument for the risk factors of developing XDR-TB from MDR-TB Reference 4 (WHO TB report 2012) should be updated Reference 35 has no Journal name Reference 37 has no title Reference 38 contains a printing error ("resistant. Tuberculosis". suppress the "." 

Author Response

Short-course regimen for multidrug-resistant tuberculosis: a decade of evidence

Reviewer N°1

This is a good review of the evidence of the efficacy of the short-course regimen (SCR) for the treatment of multidrug-resistant tuberculosis (MDR-TB), including an injectable second-line drug, as opposed to the new all-oral treatment proposed by WHO. The authors present the arguments for keeping the SCR as a cheap and efficient option, particularly in countries with limited resources. Furthermore, they present the evidence demonstrating that gatifloxacin is the most efficient quinolone and should be re-introduced on the market.

An interesting point is also the fact that, according to the authors, the presence of resistance against some of the companion drugs (pyrazinamide, thioamides and ethambutol) has no impact on the final outcome of treatment. This is in contradiction with the current position of WHO, which proposes the new oral treatment as a method for overcoming these additional resistances and as an argument against the SCR.

Main comments:

Question 1. The main issue is the fact that there is no consensus or evidence for the use of SCR in regions with the highest proportion of MDR-TB, namely Eastern Europe, where additional resistance to several second-line drugs is current and the unabated transmission of resistant strains represents a major threat for public health. The authors use the evidence from the meta-analysis of Ahuja (2012) for defending the use of injectables, ethambutol, thioamides and pyrazinamide but do not consider the fact that, in this analysis, a large proportion of patients did not receive quinolones and the results probably do not represent the current reality.

Response 1. We agree with the reviewer that there is no evidence on the effectiveness of the SCR in patients with FLQ susceptible TB in Eastern Europe because the SCR has rarely been used in Eastern Europe. However, there is also no evidence that the long all oral WHO recommended regimen would work better.

The use of of injectables, ethambutol, thioamides and pyrazinamide in the SCR was not based on finding of Ahuja; the roles of these drugs in the SCR have been discussed previously (Eur Respir J. 2017;49:1700223). One of the limitatiosn of the cited meta-analyses are that they study the effect of individual drugs, ignoring the regimen in which these drugs were used. Therefore, the reason why some results from one meta-analysis are not confirmed by the follow-up meta-analysis. In the manuscript we don’t argue that one or other meta-analysis is correct. We only alert the reader that findings from the cited meta-analysis do not seem reproducible, and should thus be interpreted with caution.

Question 2. The other point which merits a more detailed discussion is the position of injectable drugs in the treatment regimens. It would be of interest to know the opinion of the authors about the recent analysis by Reuter A et al (IJTLD 2017;21(11):1114) listing all arguments against the use of injectable drugs and the deficient evidence for their efficacy.

The role of the injectables is mainly protection against acquired resistance to the driving drug, in STR and classical long regimens for MDR-TB this is the FQ. They are strongly bactericidal early in treatment because of their action on protein metabolism, like rifampicin not having to wait for cell division and interference with cell wall building. They are active only in alkaline environment, typically near the cavity walls where the TB bacilli are most actively dividing with risk of resistance mutations coming up. Pyrazinamide action is in acid environment and intracellularly, complementing the injectables.

Important EBA was indeed never demonstrated experimentally, as pointed out by Reuter et al,, almost absent for amika and weak for strepto. This must be an artefact because they do select for resistance to themselves as soon as the driving drug is no longer actively eradicating mutants coming up against the injectable. Strepto resistance developed in about half of the patients in the early days when it was used as monotherapy. A few percent of acquired resistance to kana was recorded in STREAM and the W. African study where it was used with the poorest 4th generation FQ, moxifloxacin, also proof of selection pressure and thus activity of the drug. In STR based on the stronger gatifloxacin, acquired resistance to kana was seen in only 1 of about 500 cases on the standard regimen with at least 4 months kana, this patient had bacilli baseline susceptible only to this drug and probably CFZ. A Bangladesh gatifloxacin trial cohort, that was orally reported at Hyderabad, used only 2 months of kana to reduce toxicity. Result: as usual good cure rate of 84% for the enrolled 56 cases, but with 3 failures and 2 relapses (9%), all from baseline FQ susceptible who had acquired FQ resistance. The Reuter paper does not talk about this very important role of the injectables, i.e. protection of the driving drug against acquired resistance and treatment failure. As for the Bangladesh 2-month kanamycin cohort, the 2018 Ahmad et al. meta-analysis did not see any effect on treatment success, and missed the true unique activity of the injectables because they didn't analyse their effect on acquired resistance to the driving drug.. Additional evidence, not mentioned in the Reuter meta-analysis, comes from a large CDC Atlanta multi-country study about the determinants of acquired resistance to core drugs and evolution to XDR.1       Cegielski JP, Dalton T, Yagui T, et al. Extensive drug resistance acquired during treatment of multidrug-resistant tuberculosis. Clin Infect Dis 2014; 59: 1049-63.

It shows a considerable difference in acquired XDR and FQ resistance between patients initially susceptible to the injectables (same effect for kana, amika and capreo) compared to those resistant. In Tables 1 and 2 (37% of 226 kana resistant acquired FQ resistance, against 4% of 1049 susceptible). Reuter et al. argue that EBA has been shown for bedaquiline as well as delamanid. From the above, to us this may not at all be proof of equivalence with the injectables as main protection of the driving drug. In the case of bedaquiline least of all, because its action likely starts too late to prevent selection when the mass of bacilli and frequency of upcoming mutants is greatest. As long as this has not been cleared up, it is not justified to replace the injectables by these new drugs as high rates of acquired resistance to the driving drug will render these drugs useless very soon. Moreover, Arm C of the STREAM II clinical trial where bedaquiline replaces the injectable, compared with Arm B (classical STR but not with gati as FQ) and arm D bedaquiline replacing prothionamide and ethambutol as companion drugs) will bring the answer soon now.

As the authors rightly mention, there is currently only weak evidence of the efficacy of injection-free regimens for the treatment of MDR-TB and the included drugs are much more expensive. If, as can be expected, the evidence comes and the price of bedaquiline, linezolid and delamanid sinks, would the authors still claim that the addition of an injectable drug is necessary?

The price is not the main argument for the use of SCR. We argue that bedaquiline is an excellent drug with high effectiveness. Therefore, we are convinced that it should be safeguarded to be used as core drug of a treatment regimen for those patients who fail the 9-month regimen or present with fluoroquinolone-resistant TB. If we would use it widespread in all patients with rifampicin-resistant TB, despite the current very effective SCR, we take the risk to lose it quite quickly. Linezolid is very effective but unfortunately very toxic; in the last data from the NIX study, > 70% of their patients had to stop Linezolid for a while or definitively. Therefore, we think that linezolid should not be used widespread, but only for specific indications, like XDR-TB or in a modified SCR in patients with ototoxicity. The bacteriological activity of Delamanid is yet unclear. To our knowledge, it has not yet been tested to replace the injectable in a 9-month regimen. If evidence would show that an oral drug can replace the injectable, without compromising the treatment of those who fail the 9-month treatment, for sure we would be very happy. It is what we say in the conclusion: “It would be a real progress to abandon the use of injectable drugs but so far there is very weak evidence for the efficacy of injection-free regimens and we recommend to wait for the results of ongoing studies.”    

Minor comments:

Fig 1: please clarify the abbreviations GFX, GFXh, MFX and MFXh in the legend  

            Done

Line 185 amplification of resistance: can probably be assessed only in patients with relapse. Please clarify.  

It can be assessed also among the patients for whom the treatment fails. It is also important only in those who may transmit these bacilli, recurrences. We added :” Amplification of resistance: can be assessed only in patients with relapse or failure”.

Line 274, discussion. Sequential addition of single drugs and insufficient dosage / serum levels of drugs with low bioavailability is probably much more important for the development of resistance than intermittent use of drugs in correct dosage. Suggest to add "..intake of anti-TB drugs in insufficient or inappropriate dose… favour the development of XDR".

            Done

I also suggest to add the reference from Shin SS, AJRCCM 2010:182:426-432 as an argument for the risk factors of developing XDR-TB from MDR-TB

Reference added, thank you

Reference 4 (WHO TB report 2012) should be updated

            Thank you for the suggestion, but we would prefer to show MDR-TB outcomes reported in 2012, which facilitates interpreting early evidence on the outcomes of the 9-month regimen.

Reference 35 has no Journal name

Omission repaired: Int J Antimicrob Agents

Reference 37 has no title

Sorry: “Acquisition of cross-resistance to Bedaquiline and Clofazimine following treatment for Tuberculosis in Pakistan”

Reference 38 contains a printing error ("resistant. Tuberculosis". suppress the "." 

            Done, thank you

Reviewer 2 Report

The review article by Trebucq provides a summary of recent clinical studies of short course regiment for MDR-TB treatment. The paper is generally well written. I have only a few minor points:

What is the follow up duration for the six studies? Are they all two years or different? This information is needed since it may affect the outcome. In table 5, what does the column ‘still susceptible’ mean? Does it mean cases of failure or non-responding caused by still susceptible strains? This should be explained or clarified.

Author Response

Short-course regimen for multidrug-resistant tuberculosis: a decade of evidence

Reviewer 2

What is the follow up duration for the six studies? Are they all two years or different? This information is needed since it may affect the outcome.

The follow-up duration was 2 years for Bangladesh and West/Central Africa studies, until week 132 for Stream, one year for Benin, Niger and Cameroon.

In table 5, what does the column ‘still susceptible’ mean? Does it mean cases of failure or non-responding caused by still susceptible strains? This should be explained or clarified.

Thank you. To make it clearer, we changed the column tittle to “Susceptible”, and we added a footnote “Patients who failed or relapsed with strains still susceptible to fluoroquinolones”